# Subclinical Target Organ Damage in a Sample of Children with Autosomal Dominant Polycystic Kidney Disease: A Pilot Study

**DOI:** 10.3390/medicina59101777

**Published:** 2023-10-05

**Authors:** Simone Romano, Denise Marcon, Lorella Branz, Angela Tagetti, Giada Monamì, Alice Giontella, Francesca Malesani, Luca Pecoraro, Pietro Minuz, Milena Brugnara, Cristiano Fava

**Affiliations:** 1Section of General Medicine and Hypertension, Department of Medicine, Policlinico GB Rossi, University of Verona, 37134 Verona, Italydenise.m@hotmail.it (D.M.); lorellabranz@yahoo.com (L.B.); alice.giontella@med.lu.se (A.G.); pietro.minuz@univr.it (P.M.); cristiano.fava@univr.it (C.F.); 2Pediatric Clinic, Department of Surgical Sciences, Dentistry, Gynecology and Pediatrics, University of Verona, 37126 Verona, Italy

**Keywords:** ADPKD, hypertension, vascular damages, cardiac remodeling, cardiac hypertrophy

## Abstract

*Background and Objectives*: Hypertension and vascular damage can begin in adolescents affected by Autosomal Dominant Polycystic Kidney Disease (ADPKD). This study aimed to evaluate markers of vascular damage and left ventricular geometry in a sample of children with ADPKD. *Materials and Methods*: Several vascular measurements were obtained: ambulatory blood pressure monitoring (ABPM), carotid intima-media thickness (cIMT), carotid distensibility coefficient (cDC), pulse wave velocity (PWV), and echocardiographic measurements (relative wall thickness (RWT) and left ventricular mass index (LVMI)). *Results*: Eleven ADPKD children were recruited (four females and seven males, mean age 9.5 ± 3.2 years). Four children were hypertensive at the ABPM, five were normotensive, and for two ABPM was not available. RWT was tendentially high (mean 0.47 ± 0.39). Eight patients had concentric cardiac remodeling, while one patient had cardiac hypertrophy. cIMT was above the 95° percentile for sex and height in 80% of the children (0.5 ± 0.005 mm). The average PWV and cDC were between the normal range (5.5 ± 4.6 m/s and 89.6 ± 16.1 × 10^−3^/KPa, respectively). We observed a positive correlation between the PWV and RWT (r = 0.616; *p* = 0.044) and a negative correlation between cDC and RWT (r = −0.770; *p* = 0.015). Cardiovascular damages (cIMT > 95° percentile) were found in normotensive patients. *Conclusions*: Increased RWT and high cIMT, indicating subclinical organ damage, are already present in ADPKD children. RWT was significantly correlated to that of cDC and PWV, implying that vascular stiffening is associated with cardiac remodeling. None of the children had an alteration in renal function. Subclinical cardiovascular damage preceded the decline in glomerular filtration rate.

## 1. Introduction

Autosomal dominant polycystic kidney disease (ADPKD) is a disorder occurring in about 1 in every 400 to 1000 live births [1,2]. Most patients with ADPKD have an abnormality on chromosome 16 (PKD1 locus) [3], although other defects are reported [4,5]. In approximately eight percent of families, no mutation is detected. The disease leads to several cysts in the renal tubule, affecting its function over time. Renal ultrasonography is usually used for screening. A genetic diagnosis can be performed when a definitive diagnosis is required [6]. Usually, the renal manifestation is predominant: hematuria [7], proteinuria [8], nephrolithiasis [9], and finally renal insufficiency, although a prognostic model has been developed for identifying high-risk patients setting renal end-stage earlier [10]. Nevertheless, extra-renal manifestations such as cerebral aneurysms, hepatic and pancreatic cysts, and valvular abnormalities have also been reported [11,12,13,14]. Hypertension is another common disease present in these patients [15]. Still, although hypertension is common in most chronic progressive kidney diseases, the pathogenesis is somewhat different because it correlates to endothelial dysfunction, especially for a reduction in nitric oxide and an over-activation of the renin-angiotensin-aldosterone system [16,17,18]. Most patients with ADPKD die from cardiovascular causes [19,20,21]. An increase in arterial stiffness, left ventricular hypertrophy, and increased carotid intima-media thickness have been reported in the adult population, and therefore strictly correlated to cardiovascular events [22,23,24,25]. Childhood cardiac and vascular damage data are partial and very limited [26,27]. Identifying early cardiovascular dysfunctions during childhood, such as hypertension, cardiac remodeling or hypertrophy, arterial stiffness, and carotid intima-media thickness, could be important to prevent future cardiovascular events. Thus, the aim of this study is the evaluation of markers of early vascular damage and left ventricular geometry in a sample of children affected by ADPKD.

## 2. Materials and Methods

Patients with ADPKD were recruited from October to December 2018. These patients were referred to the Unit of Pediatric Nephrology of the University Hospital of Verona. The study was approved by the Ethical Committee of the University Hospital of Verona (CESC n.9427), and written informed consent was obtained from each participant’s parents.

Several vascular measurements were obtained. Specifically, ambulatory blood pressure monitoring (ABPM), office blood pressure (OBP), carotid intima-media thickness (cIMT), carotid distensibility (DC), pulse wave velocity (PWV), and echocardiographic measurements (relative wall thickness (RWT) and left ventricular mass index (LVMI)).

### 2.1. Blood Pressure Measurement and Vascular Exams

An oscillometric device recorded ABPM (Intermed A&D TM-2430). It was placed on the non-dominant arm and was set such that measurements were taken every 15 min during the day and every 30 min throughout the night, adapting “day” and “night” according to the diary form completed by the child or parents. All the values derived from the blood pressure (BP) measurements were also Z-score transformed according to normative values [28,29].

cIMT was assessed with an ultrasound of the carotid arteries (LogiQ P5 Pro, Bimedis, Kissimmee, FL, USA). The cIMT was estimated by tracking the artery wall in the last centimeter of the common carotid artery and calculated using dedicated hardware (Carotid studio, Quipu, Pisa, Italy). The relative z-scores and percentiles were calculated using reference values [30].

The cDC was calculated as cDC = ΔA/(A × ΔP), where A is the diastolic lumen area, ΔA is the stroke change in lumen area, and ΔP is pulse pressure (PP). Changes in diameters were detected using ultrasound B-mode image sequences of the right and left common carotid arteries acquired at different steps. These changes were analyzed using the above-mentioned automatic system [31]. The relative z-score and percentile were calculated according to reference values [32].

PWV was measured with SphygmoCor XCEL (AtCor Medical Pty Ltd.; Unit 11, West Ryde Corporate Centre, 1059–1063 Victoria Road, West Ryde, NSW 2114, Australia). To conduct a carotid–femoral PWV measurement, a cuff was placed around the femoral artery of the child to capture the femoral waveform, and a tonometer was used to capture the carotid waveform. The distance between the carotid and femoral arteries was measured, and the velocity was automatically determined by dividing the distance by the pulse transit time. The subtraction method calculated the distance between the carotid measurement site and the cuffed site. The distance was calculated from the sternal notch to the top edge of the femoral cuff (distal distance) and from the carotid artery to the sternal notch (proximal distance). To assess the above, the proximal distance was subtracted from the distal distance to determine the aortic lative z-score, and the percentile was calculated according to reference values [33].

A single sonographer performed transthoracic echocardiography (Esaote MyLabTM40, Roma, Italy) in all participants. In the parasternal long-axis view, the 2D method was used to measure interventricular septum thickness end-diastole (IVSd), left ventricular end-diastolic diameter (LVEDd), and left ventricular posterior wall thickness at end-diastole. The Relative Wall Thickness (RWT) was calculated through the following formula: (IVSd + LVPWd/LVEDd). The Devereux equation was used to obtain left ventricular mass (LVM = 0.80 * 1.04 [(tele-diastolic diameter + PW + IVS)^3^ − tele-diastolic diameter^3^] + 0.6 gr). LVM was indexed (LVMI) to height (m2.7) [34]. Left ventricular hypertrophy (LVH) was defined as the presence of an LVMI greater or equal to the 95th percentile, specific for age and sex [35]. The threshold for increased RWT (adjusted) was 0.375; the 95th percentile was specific for age [36]. Normal geometry was defined by normal LVMI and normal RWT, concentric remodeling by normal LVMI and increased RWT, concentric hypertrophy by increased LVMI and RWT, and eccentric hypertrophy by increased LVMI and normal RWT [37].

### 2.2. Laboratory Exams

Venous blood and urinary samples were collected after an overnight fast. Biochemical parameters such as triglycerides, total cholesterol, HDL cholesterol, LDL cholesterol, non-HDL cholesterol, uric acid, creatinine, cystatin C, microalbuminuria (spot urine sample), and proteinuria (24 h urine collection) were analyzed in a single reference centralized laboratory with standard methods on routine clinical chemistry instrumentation (Cobas 8000, Roche Diagnostics GmbH, Mannheim, Germany). The estimated glomerular filtration rate (eGFR) was calculated using the Schwartz equation.

### 2.3. Anthropometric Parameters

Anthropometric parameters were also assessed. Body weight was measured using a calibrated balance, and height was measured using a calibrated stadiometer. The body mass index (BMI) was calculated as weight (kg) divided by the square of height (m). Overweight and obesity were defined as BMI ≥ 85th and 95th percentile for sex and age, respectively [38]. The WHO reference for BMI categorizes children into the overweight and obese groups [39].

### 2.4. Statistical Analysis

For the statistical analysis, data were expressed as mean ± standard deviation for continuous variables or percentages for categorical ones. The Spearman correlation coefficient (rS) was used to quantify the linear relationship between variables. Statistical analyses were performed using SPSS software (IBM Corp. Released 2015. IBM SPSS Statistics for Windows, Version 23.0. Armonk, NY, USA: IBM Corp). Graphs were created with GraphPad Prism version 7.00 for Windows, GraphPad Software, La Jolla, CA, USA (www.graphpad.com).

## 3. Results

Patients’ characteristics are reported in Table 1.

Eleven patients, seven males and four females, were included in the study. The mean age of the population was 9.5 ± 3.2 years. The median BMI was 18.9 ± 3.4 kg/m^2^. Three children were overweight (>85th), and eight were normal weight. None of the subjects had altered glomerular filtration rate (GFR), and the mean GFR was 109.6 ± 13.5 mL/min/m^2^. The mean microalbuminuria measured by the albumin mg/mmol ratio was 1.1 ± 1.3. The cardiovascular parameters according to the percentile reference values are reported in Table 2.

Two ABPM, one cIMT, three cDC, and two PWV measurements were unavailable. Four children had high blood pressure (HBP) according to ABPM, five were normotensive, and only two patients were hypertensive at office blood pressure (OBP) measurement, consistent with the possible diagnosis of masked hypertension. One child was already under chronic therapy with an ACE inhibitor. RWT was, on average, higher concerning the cut-off for remodeling (mean 0.39 ± 0.05). In particular, eight patients had concentric cardiac remodeling, while one had cardiac concentric hypertrophy. cIMT was above the 95th percentile for sex and height in 80% of children (0.5 ± 0.005 mm), while average PWV and cDC were within the normal range (5.5 ± 4.6 m/s and 89.6 ± 16.1 × 10^−3^/KPa, respectively). We observed a positive correlation between the PWV and RWT (rS = 0.616; *p* = 0.044; Figure 1a) and a negative correlation between cDC and RWT (rS = −0.770; *p* = 0.015; Figure 1b), suggesting a close relationship between vascular dysfunction and initial cardiac damage.

Of note, cardiovascular damages were also found in normotensive patients; three patients already had cardiac remodeling, one had cardiac concentric hypertrophy, and four were in the IMT ≥ 95° percentile. Probably, because of the early age of our population, we found more concentric remodeling than cardiac hypertrophy. No correlations between cardiovascular parameters and either proteinuria or GFR were found.

## 4. Discussion

This study showed the prevalence of early organ damage in children affected by ADPKD. Nowak et al. [26] demonstrated vascular dysfunction measured by brachial artery flow-mediated dilation and arterial stiffness, measured as carotid–femoral pulse wave velocity. Still, it was measured in young adults with a mean age of 21. In addition, they did not assess cardiac parameters. In the only other paper conducted on children, Karava et al. [27] demonstrated in 21 adolescents a high PWV and increased cIMT in comparison with matched controls, indicating an increase in arterial stiffness and hypertrophic vasculopathy. They also found that around 10% of patients had left ventricular hypertrophy (but only in patients on antihypertensive treatment) and a linear correlation between LVH and PWV and LVH and cIMT. Patients with LVH were older, suggesting that arterial dysfunction precedes cardiac damage. Cardiac remodeling RWT was not assessed. A total of 19% of patients had hypertension. Of note, even Karava’s population was older than ours because the mean age was 12 years. Few other studies on adolescents with ADPKD showed a higher LVMI in hypertensive and borderline hypertensive adolescents than non-hypertensive [40]. A higher LVMI in patients with ADPKD as compared to group control was also shown, although within the normal range [41,42]. It has also been reported that around half of adult patients with ADPKD have LVH [43]. All the remaining studies investigating cardiac and vascular damage in patients with ADPKD were conducted in adults [14,15,17,44,45,46]. In the present study, we have shown a very early vascular and cardiac impairment in a patient affected by ADPKD despite a mean age of only 9.5 years. We have found neither PWV nor DC beyond the reference range, but different from the study of Karava et al. [27], we showed that 80% of patients had a cIMT higher than the 95th percentile. Regarding cardiac damage, the mean RWT was increased on average, and this was a sign of cardiac remodeling, whereas only one child had cardiac concentric hypertrophy. This is probably due to the very young age of our population, with concentric remodeling being a (possible) first step before overt cardiac hypertrophy can develop. It is also intuitive that vascular dysfunction precedes cardiac damage. Finally, we found a positive correlation between PWV and RWT, as well as a negative correlation between DC and RWT. Taken together, these unexpected findings suggest that vascular and cardiac damage is already detectable in children <10 years and that they are in part related to each other. This evidence is in line with the fact that most ADPKD patients die because of cardiovascular diseases [19,47]. Despite the evidence of cardiac and vascular damage, only two patients out of eleven in our sample could be classified as hypertensive if evaluated using office BP. Conversely, if we refer to ABPM, the number of hypertensive children rises to four out of ten, compatible with a diagnosis of masked hypertension. This suggests that hypertension is underestimated, and a widespread use of ABPM should probably be advised as a better screening tool in all patients affected by ADPKD. In previous studies using office BP, the median age for hypertension diagnosis in patients with ADPKD was 33 years in males and 38 years in females whose parents were hypertensive and 40 years in males and 50 years in females whose parents were non-hypertensive [48]. Interestingly, in our sample, target organ damage was not found only in hypertensive children but also in some normotensive ones. This is a significant finding, suggesting that cardiovascular injuries in these patients are not only related to hypertension. This hypothesis is consistent with the fact that the damage caused by hypertension takes several years. Due to the young age of our population, one can speculate that other mechanisms can subtend cardiovascular damage besides and beyond hypertension per se.

The reason why patients with ADPKD develop so early and so frequently experience cardiovascular damage is still debated. The activation of the renin-angiotensin-aldosterone-system (RAAS) [17,18,49,50,51,52], an increased sympathetic tone [53], and endothelial dysfunction, primarily via endothelin and nitric oxide actions [16,54,55,56,57,58,59], were intensively investigated.

According to the results of our study, the role of blood pressure/hypertension is not pivotal since the markers of early vascular and cardiac damage are not different in children with or without high blood pressure (Table 1), and cardiovascular damage was present in normotensive patients.

Anyhow, it is conceivable to speculate that early antihypertensive treatment, especially with drugs that modulate renin-angiotensin-aldosterone, may be important for preventing renal worsening or cardiovascular protection. Both these speculations are supported by the notion that cardiovascular disease prevention occurs by reducing blood pressure and LVH, even in ADPKD patients [60,61]. So far, the use of ACE inhibitors and maintaining blood pressure < 120/80 mmHg are recommended in hypertensive patients with ADPKD [62]. By contrast, if ACE inhibitors have been shown to slow kidney disease progression in patients with nephropathies, especially if proteinuria is present, the role of ACE inhibitors in slowing the progression of ADPKD needs to be clarified. In a randomized, prospective trial, after a 5-year follow-up in patients with ADPKD who had well-preserved renal function, amlodipine and enalapril were associated with a similar decline in creatinine clearance. Still, only enalapril showed a sustained antialbuminuric effect leading to higher protective effects in the long term [63]. No differences in the loss of renal function were also found in comparison with beta-blockers after a 3-year follow-up [64]. On the other hand, it has been demonstrated that lowering blood pressure is crucial in slowing renal damage progression regardless of antihypertensive use [64,65]. Even a meta-analysis including 142 patients affected by ADPKD concluded that ACE inhibitors are more effective in lowering urine protein excretion. Still, the progression rate of kidney disease was not significantly slower than other agents [62]. All these data are in line with the hypothesis that, for slowing kidney disease progression in ADPKD, lowering blood pressure is particularly important, regardless of the therapy used, even though a more extended trial may be needed.

The cross-sectional design of the present study does not allow us to see if cardiac and vascular damages develop contemporarily or are, in part, causes or consequences of each other. Another limitation of the present study is that our population consisted of only 11 patients. However, the rarity of the disease can, at least partially, justify the restricted number of included patients, which is consistent with other studies [26,27,41]. We also have to acknowledge the lack of a control group. Reference values were taken from previous epidemiological studies on children from European or US populations [32,33,34,35,36]. Moreover, another limit was that genetics data were not available.

## 5. Conclusions

In conclusion, in this study, we showed a high prevalence of organ damage even in children (with an average age < 10 years old) affected by ADPKD. In these children, we report tendentially increased RWT and high cIMT, with likely masked hypertension in almost half of patients despite normal renal function. Although this observation is clinically relevant, the present study should be read cautiously due to several limitations. First of all, because of the small sample size, this remains a pilot study, and firm conclusions cannot be drawn. Second, a control group is missing, and thus differences between ADPKD patients and healthy children cannot be determined, but only interpreted in the light of references values. Anyhow, our data should be a warning and prompt towards early cardiovascular screening and BP measurements using ABPM in children affected by ADPKD. Future studies using a higher sample size and adequate control groups may help to address the unmet points raised by our pilot study.

## Figures and Tables

**Figure 1 medicina-59-01777-f001:**
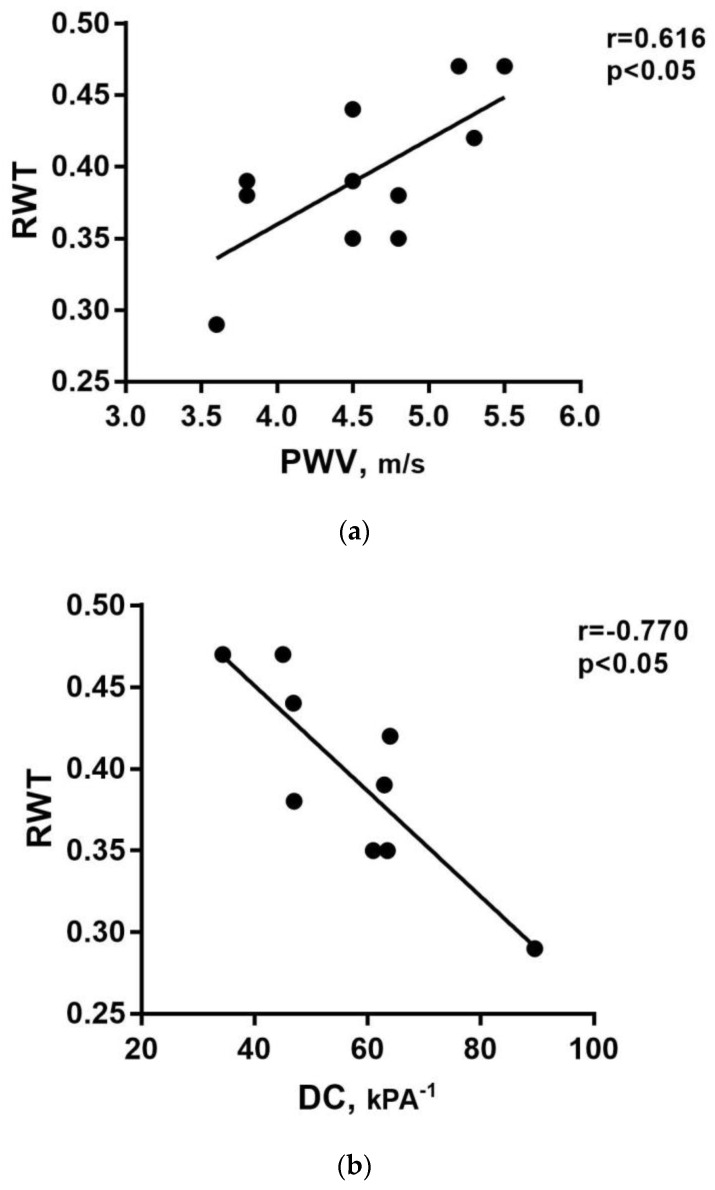
(**a**) Linear positive correlation between the pulse wave velocity (PWV) and relative wall thickness (RWT). (**b**) Linear negative correlation between carotid distensibility coefficient (cDC) and RWT.

**Table 1 medicina-59-01777-t001:** ADPKD patients’ characteristics stratified by sex and high versus normal blood pressure.

	Female	Male		High BP at ABPM/OBP	Normal BP at ABPM/OBP	
	(*n* = 4)	(*n* = 7)		(*n* = 5)	(*n* = 6)	
	Mean ± SD	Mean ± SD	*p*-Value	Mean ± SD	Mean ± SD	*p*-Value
Age, years	8.8 ± 3.4	9.9 ± 3.3	0.612	8.6 ± 2.6	10.2 ± 3.8	0.453
BMI, kg/m^2^	19.8 ± 2.8	18.4 ± 3.8	0.544	18.1 ± 3.0	19.7 ± 3.8	0.473
Percentile BMI for age	84 ± 12.6	52.8 ± 27.4	**0.030**	65.2 ± 34.9	63.3 ± 22.6	0.913
Office SBP, mmHg	110.0 ± 13.6	105.3 ± 14.5	0.610	107.8 ± 11.1	106.3 ± 16.6	0.870
Percentile SBP, mmHg	76.6 ± 23.1	59.5 ± 28	0.330	72.0 ± 19.7	60.4 ± 32.0	0.500
Office DBP, mmHg	70.3 ± 6.2	59.4 ± 8.2	**0.039**	67.6 ± 7.0	59.8 ± 9.5	0.166
Percentile DBP, mmHg	77.2 ± 13.9	39.9 ± 13	**0.005**	62.7 ± 24.8	45.7 ± 19.5	0.233
24 h SBP, mmHg	116 ± 3.6	119.2 ± 6.1	0.384	119.6 ± 7.2	116.2 ± 1.8	0.337
Percentile 24 h-SBP	83.8 ± 16.9	82.2 ± 15.9	0.880	88.0 ± 21.2	81.1 ± 4.9	0.302
24 h DBP, mmHg	66.5 ± 3.3	67.8 ± 3.4	0.559	67.8 ± 4.1	66.8 ± 2.6	0.656
Percentile 24 h-DBP	53.7 ± 27.3	54.9 ± 22.8	0.941	58.9 ± 28.0	49.9 ± 19.2	0.573
cIMT, mm	0.48 ± 0.05	0.49 ± 0.05	0.675	0.48 ± 0.05	0.49 ± 0.04	0.702
Percentile c-IMT for height	97.1 ± 3.4	96.5 ± 5.4	0.849	97.0 ± 3.0	96.4 ± 6.0	0.848
cDC, kPA-1	52.3 ± 9.2	59.6 ± 18.9	0.556	54.6 ± 9.4	51.2 ± 20.7	0.701
Percentile cDC for height	26.7 ± 14	44.7 ± 32	0.411	35.0 ± 17.5	45.3 ± 36.0	0.349
PWV, m/s	4.4 ± 0.4	4.7 ± 0.7	0.525	4.6 ± 0.61	4.5 ± 0.71	0.835
Percentile PWV for height	31.8 ± 17.2	59.5 ± 31.9	0.156	44.4 ± 19.3	52.5 ± 39.2	0.694
RWT	0.40 ± 0.03	0.39 ± 0.07	0.869	0.39 ± 0.05	0.39 ± 0.07	0.985
LVM/BSA, g/m^2^	25.8 ± 4.8	33.9 ± 9.9	0.102	27.4 ± 6.1	34.0 ± 10.6	0.249
HDL cholesterol, mg/dL	54.3 ± 8.8	55.1 ± 8.0	0.867	52.2 ± 8.9	57.0 ± 6.9	0.340
LDL cholesterol, mg/dL	73.0 ± 12.9	57.3 ± 18.4	0.171	74.7 ± 13.3	53.3 ± 15.5	**0.036**
Triglycerides, mg/dL	86.3 ± 17.6	76.3 ± 64.7	0.774	72.4 ± 24.8	86.2 ± 0.09	0.681
Uric Acid (mg/dL)	3.8 ± 1.0	3.0 ± 0.7	0.147	3.2 ± 1.2	3.4 ± 0.6	0.832
Albumin/creatinine, mg/mmoL creat.)	1.8 ± 1.9	0.56 ± 0.15	0.143	0.17 ± 2.0	0.62 ± 0.23	0.176
Schwartz eGFR, ml/min	110.8 ± 9.9	109.0 ± 16.0	0.849	119 ± 13.4	101.8 ± 7.1	**0.026**
Cystatin-c, mg/L	0.85 ± 0.05	0.87 ± 0.10	0.707	0.86 ± 0.09	0.86 ± 0.09	0.921
Proteinuria, g/die	0.07 ± 0.04	0.11 ± 0.06	0.067	75.6 ± 3.4	72.6 ± 4.7	0.277

BMI, body mass index; SBP, systolic blood pressure; DBP, diastolic blood pressure; cIMT, carotid intima-media thickness; cDC, carotid distensibility coefficient: PWV, pulse wave velocity; RWT, relative wall thickness; LVM, left ventricular mass; BSA, body surface area; HDL, high density lipoprotein; LDL, low density lipoprotein.

**Table 2 medicina-59-01777-t002:** Patients’ cardiovascular parameters stratified according to the specific 95th percentile.

	Total Population (*n* = 11)
	*n* (%)
Percentile BMI for age < 95th	8 (72.7%)
90th ≤ Percentile BMI for age < 95th	3 (27.3%)
Percentile BMI for age ≥ 95th	0
Percentile Office SBP and DBP < 95th	9 (81.8%)
Percentile Office SBP or DBP ≥ 95th	2 (18.2%)
Percentile ABPM SBP and DBP < 95th	5 (55.6%)
Percentile ABPM SBP or DBP ≥ 95th	4 (44.4%)
Percentile cIMT for height < 95th	2 (20%)
Percentile cIMT for height ≥ 95th	8 (80%)
Percentile cDC for height ≥ 5th	8 (100%)
Percentile cDC for height < 5th	0
Percentile PWV for height < 95th	9 (100%)
Percentile PWV for height ≥ 95th	0
RWT < 0.375	3 (27.3%)
RWT ≥ 0.375	8 (72.7%)
Percentile LVM < 95th	10 (90.9%)
Percentile LVM ≥ 95th	1 (9.1%)

BMI, body mass index; SBP, systolic blood pressure; DBP, diastolic blood pressure; cIMT, carotid intima-media thickness; cDC, carotid distensibility coefficient; PWV, pulse wave velocity; RWT, relative wall thickness; LVM, left ventricular mass.

## Data Availability

Not applicable.

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
