# Peer review of "Subclinical Target Organ Damage in a Sample of Children with Autosomal Dominant Polycystic Kidney Disease: A Pilot Study"

_medicina, 2023, doi:10.3390/medicina59101777_

Round 1

Reviewer 1 Report

This paper attempts to verify that autosomal dominant polycystic kidney disease (ADPKD) in children is associated with subclinical vascular damage and left ventricular geometry from the perspective of several vascular and echocardiographic measurements.

The key finding of the study is that ADPKD children already have subclinical cardiovascular damage before kidney function decline. This was shown by increased RWT and high cIMT.

Authors previously published the same data with the identical title as an abstract (doi: 10.1097/01.hjh.0000572596.21012.8d). Thereat, the difference in results is confusing – in 2019: "RWT was tendentially high (mean 0.39 ± 0.47)"; in 2023, same sample: "RWT was tendentially high (mean 0.47 ± 0.39)."

Regarding the manuscript, the introduction and discussion are well written, with adequate bibliographic references.
While, to be considered for publication, the manuscript requires thorough improvement by the authors.

The methodology is not complete. There is no information about methods, kits, and analyzers for biochemical assays. The sample size seems too small. Perhaps it should be indicated in the title that this is a pilot study. How was the optimal sample size for a study determined? What data was used as the control? Text should be divided into paragraphs with subheads according to the methods.

Results are confusing. Authors showed gender differences in the parameters studied only but didn't assess the contribution of hypertension or other factors. To assess hypertension's contribution, dividing the data according to its status or conducting multiple regression analysis are recommended. For figures and tables, explanations should be given and all abbreviations should be listed at the end with their definitions. Albumin to creatinine ratio should be given in mg/g. The specific p-values should be given.

Text should be checked for the repetitions (for example, the number of patients was mentioned in methodology and in the results). Definitions for abbreviations should be given at first mention (in the abstract – no definition for CD).

The conclusion is unconvincing. The first part is a repetition of intro, and the second part is controversial and speculative. The authors found the features of cardiovascular damage in patients without hypertension, while recommending use of blood pressure control and ACE inhibitors, whose role is obvious but was not clearly shown here.

Moderate editing of English language required

Author Response

attached file

Reviewer 2 Report

Very interesting article about Cardiovascular risk in children. Not many studies are out there in pediatric ADPKD population assessing the cardiovascular risk. Results are well presented. 

Author Response

attached file

Reviewer 3 Report

Observational study with the aim to deeply study the cardiovascular damage in a too small sample of children wiht ADPKD. 

On top of the small sample size which makes very difficult to draw any conclusion, a hypertensive child is also included and some data regarding the cardiovascular study of some of the patients is missing. No correlations are shawn between blood pressure and subclinical organ damage. A control group (or two control groups) of non ADPKD patients and patients with essential hypertension would add some valuable information. 

Finally, the discussion is too long and mentions some aspects not shown in the results (p. ex the etiology of hypertension among ADPKD patients which are not tested in the study). 

Would be a much interesting study with a greater sample size and with the two control groups, as it focuses in quite young children with ADPKD. 

Author Response

attached file

Round 2

Reviewer 1 Report

The authors have addressed my concerns and comments by editing the manuscript. I have no further questions.

Reviewer 3 Report

Thank you to the authors for having improved the mansucript according to suggestions. It is an important issue not being able to widen the sample or including a control group.